# Successful Growth of TiO_2_ Nanocrystals with {001} Facets for Solar Cells

**DOI:** 10.3390/nano13050928

**Published:** 2023-03-03

**Authors:** Saif M. H. Qaid, Hamid M. Ghaithan, Huda S. Bawazir, Abrar F. Bin Ajaj, Khulod K. AlHarbi, Abdullah S. Aldwayyan

**Affiliations:** 1Department of Physics & Astronomy, College of Sciences, King Saud University, P.O. Box 2455, Riyadh 11451, Saudi Arabia; 2K. A. CARE Energy Research and Innovation Center, King Saud University, Riyadh 11451, Saudi Arabia; 3King Abdullah Institute for Nanotechnology, King Saud University, Riyadh 11451, Saudi Arabia

**Keywords:** titanium dioxide, brookite, anatase, nanocrystal, hydrothermal, DSSCs

## Abstract

The growth of nanocrystals (NCs) from metal oxide-based substrates with exposed high-energy facets is of particular importance for many important applications, such as solar cells as photoanodes due to the high reactivity of these facets. The hydrothermal method remains a current trend for the synthesis of metal oxide nanostructures in general and titanium dioxide (TiO_2_) in particular since the calcination of the resulting powder after the completion of the hydrothermal method no longer requires a high temperature. This work aims to use a rapid hydrothermal method to synthesize numerous TiO_2_-NCs, namely, TiO_2_ nanosheets (TiO_2_-NSs), TiO_2_ nanorods (TiO_2_-NRs), and nanoparticles (TiO_2_-NPs). In these ideas, a simple non-aqueous one-pot solvothermal method was employed to prepare TiO_2_-NSs using tetrabutyl titanate Ti(OBu)_4_ as a precursor and hydrofluoric acid (HF) as a morphology control agent. Ti(OBu)_4_ alone was subjected to alcoholysis in ethanol, yielding only pure nanoparticles (TiO_2_-NPs). Subsequently, in this work, the hazardous chemical HF was replaced by sodium fluoride (NaF) as a means of controlling morphology to produce TiO_2_-NRs. The latter method was required for the growth of high purity brookite TiO_2_ NRs structure, the most difficult TiO_2_ polymorph to synthesize. The fabricated components are then morphologically evaluated using equipment, such as transmission electron microscopy (TEM), high resolution transmission electron microscopy (HRTEM), electron diffraction (SAED), and X-ray diffraction (XRD). In the results, the TEM image of the developed NCs shows the presence of TiO_2_-NSs with an average side length of about 20–30 nm and a thickness of 5–7 nm. In addition, the image TEM shows TiO_2_-NRs with diameters between 10 and 20 nm and lengths between 80 and 100 nm, together with crystals of smaller size. The phase of the crystals is good, confirmed by XRD. The anatase structure, typical of TiO_2_-NS and TiO_2_-NPs, and the high-purity brookite-TiO_2_-NRs structure, were evident in the produced nanocrystals, according to XRD. SAED patterns confirm that the synthesis of high quality single crystalline TiO_2_-NSs and TiO_2_-NRs with the exposed {001} facets are the exposed facets, which have the upper and lower dominant facets, high reactivity, high surface energy, and high surface area. TiO_2_-NSs and TiO_2_-NRs could be grown, corresponding to about 80% and 85% of the {001} outer surface area in the nanocrystal, respectively.

## 1. Introduction

Due to their low cost and high efficiency among 3rd third-generation photovoltaic technologies, dye-sensitized solar cells (DSSCs) and perovskite solar cells (PSCs) have been widely studied over the last two decades. A dye-sensitized semiconductor electrode, a counter electrode, and a redox electrolyte are typical components of DSSCs, while the same things are found in the PSCs by replacing the dye with the perovskite materials [1,2,3,4,5,6]. In these devices, a dye/perovskite sensitizer absorbs solar radiation, whereupon electrons excited by the light are transferred to the electrode, which is a wide bandgap semiconductor composed of a mesoporous oxide layer composed of nanometer-sized particles, usually titanium dioxide (TiO_2_) [7,8].

TiO_2_ is used as a photoanode in both DSSCs and PSCs, which has a significant effect on the film thickness and photovoltaic performance [5,6,9]. The reflectance spectra from ultraviolet to visible region, conversion efficiency of incident monochromatic photons into current, and electrochemical impedance spectroscopy measurements can be used to measure the photoelectrochemical properties of photoanodes. Due to their large surface area and charge transport properties, which also exhibit a high electron transport rate, photoanodes are considered superior electrode designs for the development of photovoltaic devices. By creating filigree hierarchical TiO_2_ nanostructures, research has significantly improved the power conversion efficiency (PCE) [10]. However, the improvement of TiO_2_-based DSSCs is limited by the low absorption coefficient of conventional organic dyes, whereas the high absorption coefficient of perovskite materials makes them very promising in the near future in near future.

There are three crystalline forms of titanium dioxide are present in nature: anatase, rutile, and brookite [11]. TiO_2_ is widely used as a white pigment and sunscreen. It is also an attractive material for numerous thin film applications because it is chemically stable (good chemical and abrasion resistance), abundant, inexpensive, nontoxic, and readily available in large quantities. It has a high dielectric constant, which makes it desirable as a dielectric material in the electronics industry [9,12,13]. For this reason, the fabrication of metallic dielectric semiconductor structures with different dielectric gates is possible. In addition, it possesses several properties, such as a high refractive index with excellent transmittance in visible and near infrared wavelength regions [2]. Due to its large band gap of 3.2, 3.0, and 2.95 eV for anatase, rutile, and brookite, respectively, it can absorb several percentages of sunlight in the near-UV region. Therefore, the theoretical maximum solar energy conversion efficiency (AM 1.5 G) is 2.2% [1,2,3]. As for the three crystalline phases of TiO_2_, rutile is most stable in the bulk phase at high temperatures and ambient pressure. The key to fostering future progress in all promising technological areas that benefit from nanosized TiO_2_ lies in the ability to develop nanostructures that meet many requirements, such as defined crystal phases and degrees of crystallinity, tailored dimensions and shapes, and appropriate chemical functionalities at the surface [14]. Numerous studies have examined the formation of anatase TiO_2_ nanostructures with various morphologies because the anatase phase is more stable at the nanoscale [15,16]. Under certain growth conditions, stable nanoscale rutile and brookite nanostructures can also be generated [16,17,18].

The brookite phase has gotten the least attention and is the hardest to synthesis when compared to rutile and anatase [19].

In contrast to rutile and anatase TiO_2_, the main challenges in synthesizing brookite NCs are the complexity and special requirements needed to produce a large amount of brookite phase, including a specific precursor, pH, and additive. In general, crystalline anatase phase exhibits much higher photocatalytic activity than rutile and brookite [20,21]. It can be produced in large quantities in aerosol flame reactors using vaporous organotitanium or titanium chloride precursors. Sodium titanate was used as a precursor in a simple hydrothermal process to produce brookite TiO_2_, which was then produced in the presence of an aqueous sodium fluoride solution. NaF concentration can be used to change the ratio of brookite and anatase TiO_2_ [22]. We obtained high quality brookite TiO_2_ in a concentrated NaF solution. The nominal atomic ratio of fluorine to titanium was 1. Fluorine can change the particle morphology of TiO_2_ and improve the photocatalytic activity. Pure anatase nanoparticles were obtained only in deionized water. In addition, the morphology and size of brookite TiO_2_ can be tailored by using different acid-treated titanates that affect the stability of the building blocks and nucleation points of TiO_2_. Additionally, there have been numerous methods published for creating brookite nanostructures in aqueous environments [14,22,23,24,25,26,27,28,29,30,31,32], the majority of which demand protracted development durations and multiple processes. Additionally, some of the methods thus far described result in multiphase nanostructures with mixed brookite/anatase or brookite/rutile TiO_2_ phases, where the phase ratios and the consequent morphologies can be carefully regulated [14,22,23,24,25,26,27,28,29,30,31,32]. The impacts of growth factors including complex ligands, pH, and temperature have been thoroughly researched in the literature [23,26,28], and water-soluble complexes are typically utilized as titanium precursors. For instance, numerous authors have noted that strongly basic circumstances encourage the brookite phase [23,24]. When water-soluble Ti complexes are swapped out for TiCl_4_, brookite development under acidic circumstances has also been documented [29,30,31,32]. As a result, various scenarios outlining the potential mechanisms of brookite crystal development have been related to the impacts of pH and other parameters on the growth mechanism [22,24,28,29,30]. In fact, a water-based growing environment served as the common denominator in the majority of studies on the formation of TiO_2_ brookite nanostructures. Non-hydrolytic growth methods have been shown to be extremely effective for the synthesis of several oxide/mixed oxide nanostructures, allowing streamlined synthesis processes and improved control over the breakdown rates of the reaction precursors [33,34]. To our knowledge, there are not many studies on brookite TiO_2_ development in non-hydrolytic settings. By using high temperature aminolysis of titanium carboxylate complexes, Buonsanti et al. [14,15] demonstrated a sophisticated non-aqueous surfactant assisted approach for the manufacture of pure brookite nanocrystals. Lu and coworkers reported the synthesis of anatase TiO_2_ microcrystals with high-energy {001} facets by using hydrofluoric acid as a shape-controlling agent [35]. Han and co-workers used a similar strategy to synthesize anatase TiO_2_ nanosheets with 89% exposed {001} facets using a hydrofluoric acid solution as a solvent [36]. Since the (001) surface of anatase TiO_2_ nanosheets is much more reactive than the thermodynamically stable (101) surface, the obtained TiO_2_ nanosheets provide a new and good opportunity to develop photocatalytic materials and devices with higher activity. The manufacture of DSSCs based on two-dimensional anatase TiO_2_-NSs with exposed 001 facets and their photoelectric conversion efficiencies, however, has received very little attention, as far as we are aware. The anatase and brookite phases are low pressure and low temperature forms. Rutile and anatase are both tetragonal, while brookite is orthorhombic [37]. Based on the above discussion on the preparation and applications of TiO_2_-NCs, the hydrothermal method is still a current trend for the preparation of metal oxide nanostructures in general and TiO_2_ in particular, since the calcination of the resulting powder after the completion of the hydrothermal method no longer requires a high temperature. This can prevent the nanostructures from clumping and contaminating again. In addition, the operating temperature, pressure, and reaction path during the process of producing NCs by hydrothermal methods affect the distribution, shape, and size of the particles formed. For the preparation of TiO_2_-NCs, TiO_2_ precursor and fluorine precursor are used as morphology control agents.

Here, tetrabutyl titanate Ti(OBu)_4_ served as the precursor for TiO_2_ in each experiment. As morphological regulating agents, hydrofluoric acid (HF) and sodium fluoride (NaF) are used in a straightforward non-aqueous one-pot solvothermal method to produce TiO_2_-NSs and TiO_2_-NRs, respectively. The direct alcoholysis of Ti(OBu)_4_ in ethanol in the presence of hydrofluoric acid (HF) and sodium fluoride (NaF) salt, respectively, yields high-quality anatase nanosheets and brookite nanorods in a completely non-aqueous environment. However, alcoholysis of only TiO_2_ precursor in ethanol without fluorine precursor resulted in pure nanoparticles (TiO_2_-NPs) in the absence of a morphology control agent. Subsequently, TiO_2_ morphology, porous structure, and crystallinity play an important role in the photoelectric conversion efficiency of DSSCs, so we use fluorine, which can enhance the photocatalytic activity and control the particle morphology of TiO_2_. The structure and morphology of the TiO_2_-NCs samples grown by the hydrothermal method were investigated using the main TEM, HRTEM, and XRD instruments.

## 2. Materials and Methods

### 2.1. Materials

Titanium butoxide (TNBT) or Ti(OBu)_4_ (Bu = CH_2_CH_2_CH_2_CH_3_) (Molar mass = 340.32 g/mol) (97%), sodium fluoride (NaF) (99%), hydrofluoric acid [HF] (40 wt%), and anhydrous ethanol (99.8%) were purchased from Sigma-Aldrich. All chemicals were used directly without further processing. Additionally, Teflon-lined stainless-steel autoclave (Parr Instrument Co., Moline, IL, USA) with a capacity of 45 mL was used.

### 2.2. Methods

#### 2.2.1. Preparation of Various TiO_2_ Nanocrystals by Hydrothermal Method

In this work, various NCs structures and morphologies of TiO_2_ were prepared by hydrothermal method. Hydrothermal synthesis is usually carried out in steel pressure vessels called autoclaves, with or without Teflon lining, under controlled temperature or pressure, where the reaction takes place in aqueous solutions. The temperature can be raised above the boiling point of water so that the pressure of steam saturation is reached. The internal pressure produced is largely determined by the temperature and the amount of solution supplied to the autoclave. This method is widely used in the ceramics industry to produce small particles [38]. Pure anatase TiO_2_-NPs were produced by alcoholysing TNBT alone in ethanol, whereas anatase TiO_2_-NS and high-purity brookite TiO_2_-NRs required the addition of either HF and NaF, respectively. The possible growth mechanism is discussed accordingly. The mixture of TiO_2_ precursor and a morphology control agent is then placed in an electric oven at 180 °C for 24 h. The atomic ratio of fluorine to titanium (F:Ti) was maintained at 1:1 for the reaction described above. After completion of the solvothermal reaction, the autoclave was allowed to cool naturally to room temperature. The white, single-crystalline TiO_2_ NCs precipitate was collected, washed three times with ethanol and distilled water, and separated by high-speed centrifugation. It was then dried in an oven at 60 °C for about 6 h [39] without the need to calcined at high temperature. Figure 1 below describes the TiO_2_-NCs growth pathway for TiO_2_-NSs, TiO_2_-NRs, and TiO_2_-NPs.

#### 2.2.2. Preparation of TiO_2_ Anatase Nanosheet

Anatase TiO_2_-NSs were prepared by the solvothermal method developed by Han et al. [36]. In a typical synthesis, 1:13 mL of HF: TNBT solution was mixed through stirred vigorously for 15 min at room temperature before being introduced inside the autoclave. Using the aforementioned process, the fluorine to titanium atomic ratio (F:Ti) was maintained at 1:1, (Sample 1).

#### 2.2.3. Preparation of TiO_2_ Brookite Nanorod

By hydrolyzing TNBT (0.966 g/cm^3^) in the presence of NaF as a powder in a straightforward, non-aqueous, one-pot solvothermal method, pure brookite TiO_2_-NRs could be produced. In a typical synthesis, 17 mL of ethanol and 1:8.11 g of NaF:TNBT were combined with a Na:Ti molar ratio of 1:1 (Sample 2) or more (Sample 3), rapidly stirred for 15 min at room temperature and then added to the autoclave. Pure anatase TiO_2_-NPs were produced by alcoholysing TNBT alone in ethanol, (Sample 4).

## 3. Sample Characterization

Transmission electron microscopy (TEM) with a field emission electron microscope analysis was performed with a JEM -2100F electron microscope (JEOL, Tokyo, Japan, models JEM -1011), using a 200 kV accelerating voltage to study the morphology and size of our samples. For the TEM observations, the powder particles were dispersed in an aqueous solution under supersonic vibration for 15 min, and then a drop of the dispersed sample was placed on the carbon coated TEM grid. Additionally, X-ray diffraction (XRD) (Xpert PAnalytical MPD) was carried out and an instrument operated at 40 KV and 15 mA and running in θ/2θ mode captured X-ray diffraction spectra. At room temperature, transmission spectra of the thin films were recorded using a V-670 UV-Vis spectrophotometer (JASCO, Tokyo, Japan). The thickness of the film was measured with a surface profilometer (Veeco Dektak 150).

## 4. Results and Discussion

In this paper, some characterization for TiO_2_ nanocrystals (TiO_2_-NCs) structure prepared by hydrothermal method to use later as an electrode in the DSSC device. In a straightforward non-aqueous one-pot solvothermal method, TNBT was used as a titanium precursor, and HF and NaF as morphology controlling agents for TiO_2_-NSs and TiO_2_-NRs, respectively. TNBT alone underwent alcoholysis in ethanol, producing solely pure TiO_2_-NPs. The average yields were 43.4%, 42.6%, and 45.0% for TiO_2_-NSs, TiO_2_-NRs, and TiO_2_-NPs, respectively. Figure 2 illustrates the different TiO_2_-NCs shapes.

## 5. Morphological Properties

The first sample was synthesized using TNBT as a precursor and HF as a morphology-controlling agent. Figure 1 shows the TEM, HRTEM, and SAED images of the fabricated TiO_2_-NSs to characterize the size and morphology of synthesized particles.

The particles formed in this experiment can be seen in the TiO_2_-NSs on the TEM image of Figure 1a, which clearly shows the crystalline nature of the TiO_2_-NSs and confirms that the synthesized powder is composed of TiO_2_-NSs. The TEM image shows that the TiO_2_-NSs are almost uniformly distributed and have a wide size distribution, with a minimum size of about 5 nm. As shown in Figure 1a, the TEM images confirm that the fabricated samples consist of well-defined sheet-like structures with a rectangular outline with an average side size of ~20–30 nm wide and 7–10 nm thick. The TEM image implies that the NSs have uniform distribution and good dispersion, which makes them promising for solar cell applications with large surface area. The single crystalline nature of the TiO_2_-NSs can be seen from the HRTEM image in Figure 1b, where the lattice planes can be seen. The lattice spacing in Figure 1b was measured to be ~0.35 nm, which corresponds to the {001} planes of anatase TiO_2_, indicating that the top and bottom facets of the NSs are the {001} and {001} planes, respectively. This corresponded to the distance between adjacent {001} lattice planes, which was confirmed later by performing an XRD analysis. SAED was performed for a planar nanosheet, and the result is shown in Figure 1c. The diffraction pattern also verifies that our nanosheets are solitary crystals. Additionally, the anatase crystal structure’s zone axis along the {001} orientation could be used to index the SAED pattern. This confirms that the exposed surfaces of the nanosheets at the top and bottom are the {001} facets. The growth of TiO_2_ nanosheets requires the use of HF acid, and we are trying to develop other methods to replace this dangerous chemical. The choice and optimization of the proper coating method and parameters of the ETM layer of TiO_2_ nanosheets still need to be optimized to achieve the best efficiency, as explained in the methodology section. Based on the TEM and HRTEM results, we can roughly calculate the percentage of exposed {001} facets on the TiO_2_-NSs. The percentage of exposed {001} facets representing ~80% of the {001} outer surface area in the NCs. HF is extremely corrosive and contact poison. So, it should be handled with great caution. Additionally, HF acid solution is kept in active Teflon receptacles. Therefore, the second sample was synthesized using TNBT as a precursor and NaF as a morphology control agent for more safety. The TEM image and SAED of the TiO_2_-NCs sample in this condition were performed. We also report the synthesis of anatase TiO_2_ quantum dots with a size of about 5 nm in combination with brookite TiO_2_ nanorods (Figure 2).

In addition, the image TEM shows TiO_2_-NRs with diameters between 10 and 20 nm and lengths between 80 and 100 nm, together with crystals of smaller size. Thus, the synthesis method needs further improvement to increase the growth selectivity during the synthesis process. It is essential to note that the NaF concentration played a significant role and that the pure brookite phase could only be produced when the Na:Ti molar ratio was at least 1:1. High-quality brookite TiO_2_-NRs were prepared in a concentrated NaF solution. Pure anatase TiO_2_-NPs were prepared without the use of NaF. Moreover, since NaF concentration affects the morphology and size of TiO_2_-NCs, it is possible to change the ratio of brookite and anatase TiO_2_ [22]. Fluorine is believed to play an important role in how the morphology and size of TiO_2_ are affected by NaF concentration, as it can increase the photocatalytic activity and regulate the morphology of TiO_2_ particles. To successfully make a pure brookite phase, the third sample was created using TNBT as a precursor and NaF as a morphology control agent. TEM and HRTEM pictures of the created TiO_2_ nanorods are shown in Figure 3. (TiO_2_-NRs). The nanorods have average lateral dimensions of ~12 nm and ~80 nm, while the nanoparticles have average lateral sizes of ~5 nm. HRTEM images confirm the anatase and brookite phases in both samples. The HRTEM image of the brookite sample (Figure 3b) shows the atomic levels. Figure 3b shows the ImageJ-determined lattice fringes of a single NR with a lattice plane spacing of ~0.25 nm, which corresponds to the 101-lattice spacing. This corresponded to the spacing between adjacent (200) lattice planes, which was confirmed by XRD analysis. The SAED pattern corresponds to the diffraction pattern of anatase TiO_2_ with the zone axis along the {121} direction [15].

Our findings unequivocally support the brookite phase of the nanorods because anatase and rutile formations don’t have the same lattice spacing. The small crystals also exhibit brookite phase, which was later confirmed by XRD. Although the thickness of the nanorods is small, the {101} facet accounts for up to 80% of the total thickness because it is not perpendicular to the basal planes. Due to the high-energy {101} facets, brookite nanorods are expected to be reactive in catalytic reactions, and the photocatalytic efficiency can be enhanced by trapping photoinduced holes and electrons through these redox sites, which is of great interest [30]. The fourth sample was synthesized as mentioned above, where alcoholysis of TNBT only in ethanol without morphology agent resulted in pure nanoparticles (TiO_2_-NPs). Figure 4 shows TEM images of the fabricated TiO_2_-NPs. The nanoparticles have an average size of ~5–10 nm.

In general, the HRTEM images of the samples show the atomic planes confirm with the XRD pattern the anatase and brookite phases are achieved. The brookite sample exhibits a mixture of nanorods and nanoparticles, such as QDs (Sample 2), while the anatase sample consists of small agglomerated nanoparticles of size ~5–10 nm (Sample 4).

## 6. Structural Properties

The phase structure, crystallite size, and crystallinity of TiO_2_ have a great influence on the photoelectric conversion efficiency of DSSCs. The crystalline structural properties of samples will be investigated by X-ray diffraction (XRD) technique. XRD is generally used to study the crystallinity, crystal structures, and lattice constants of the nanostructures. Here, about 10 mg of TiO_2_-NCs was pressed into a smooth layer on the surface of a metal or glass holder.

The XRD pattern of sample 1 (TiO_2_-NSs), as shown in Figure 5, contained strong, sharp, and distinct peaks at (2θ) 25.59°, 37.36°, 38.11°, 38.59°, 48.31°, 54.18°, 55.32°, 62.90°, 69.90°, 70.51°, 75.42°, and 76.27°, which correspond to diffraction from crystal planes (101), (103), (004), (112), (200), (105), (211), (204), (116), (220), (215), and (301), respectively. All peaks were assigned to the tetragonal phase [37] with space group (I4_1_/amd) as cataloged in (JCPDS card no. 21-1272). Additionally, all peaks could be assigned to the diffraction peaks of pure anatase TiO_2_. These results are consistent with those reported by others [36,39,40,41]. Due to the random distribution of nanosheets in the powder sample, all anatase peaks were observed with intensities comparable to the diffraction spectrum of the powder. Therefore, as previously stated, HRTEM and SAED are needed to confirm the single crystalline nature of the nanosheets and to identify the lattice planes along the exposed facets (Figure 1c).

The average grain size of TiO_2_-NCs was estimated using the Scherrer Equation (1) [4,42].
(1)$$d=\frac{0.9\;\lambda }{\beta \cos\theta }$$
where β is the half-full width, θ is the Bragg angle, and λ = 0.154 nm. X-ray diffraction (XRD) patterns of the grown nanosheets show a typical anatase structure. The grain size (~8–20 nm) was estimated to be ~14.73 nm using the Sherrer formula, which is consistent with observations at TEM.

In addition, a modified version of the Williamson-Hall (W-H) analysis, namely, the uniform deformation model, was used to estimate the average grain size and strain of the films. This model is determined by the following equation [43,44,45,46]:(2)$$\beta_{hkl}\cos\theta =\frac{k\lambda }{D}+4\varepsilon \sin\theta$$
where, $\beta_{hkl}$ is the peak broadening, λ is the incident radiation wavelength, ε is the strain, D is the crystallite size, and k is a constant (~0.9). The dislocation density (δ) is defined by $\delta =\frac{1}{D^{2}}$. Figure 6 shows the plot of 4sinθ versus. $\beta_{hkl}\cos\theta$, which gives a straight line with slope and intercept of ε and $\frac{k\lambda }{D}$, respectively, for the different TiO_2_-NCs (see Table 1). The results show a decrease in grain size and an increase in dislocation density and lattice strain. The determined grain size trends from XRD patterns concurred with that revealed by TEM analysis.

The linear regression equation for the data in Figure 6 was provided as follows: y=0.00718+0.00152 x. Table 1 lists the average crystalline sizes and relative crystallinity of the TiO_2_-NS samples.

We also report the synthesis of anatase crystallites TiO_2_ quantum with a size of ~5 nm mixed with brookite nanorods (second sample), as shown in Figure 7, by referring to what is cataloged in (JCPDS card no. 21-1272) and (JCPDS card no. 29-1360) for anatase and brookite, respectively.

Mixing with brookite TiO_2_ nanorods requires further enhancement of growth selectivity during the synthesis process, as in the third sample. First, the XRD pattern of the TiO_2_-NRs synthesized TNBT as a precursor and NaF as a morphology control agent (sample 3) (Figure 8) contained strong, sharp, and distinct peaks at (2θ) 25.50°, 25.86°, 31.09°, 36.51°, 37.48°, 38.09°, 39.12°, 40.52°, 42.71°, 46.86°, 48.25°, 49.86°, 54.58°, 55.43°, 56.34°, and 57.50°, which correspond to diffraction from crystal planes (120), (111), (121), (012), (201), (131), (040), (022), (221), (032), (231), (132), (320), (241), (151), and (113). All peaks were assigned to the orthorhombic phase with the space group (Pbca) as cataloged in (JCPDS card no. 29-1360). Additionally, all the peaks could be attributed to the diffraction peaks of pure brookite TiO_2_ (JCPDS No. 29-1360). These results are consistent with those reported by others [15,36,40,47]. Strong and sharp diffraction peaks are seen, and all peaks could be assigned to the diffraction peaks of pure brookite TiO_2_. Due to the random distribution of nanosheets in the powder sample, all anatase peaks were observed with intensities comparable to the diffraction spectrum of the powder. Therefore, to confirm the single crystalline nature of the nanosheets and to identify the lattice planes along their exposed facets, HRTEM and SAED are required as mentioned in Figure 3. From the above results, both rutile and anatase are tetragonal, while brookite is orthorhombic [37].

The grain size was estimated to be ~12 nm using the Sherrer formula, which is consistent with observations at TEM.

Figure 5 and Figure 8 show the XRD patterns of the two samples: with and without NaF. The XRD identification peaks of the two phases anatase (JCPDS No. 21-1272) and brookite (JCPDS No. 29-1360) are also shown in the figures. The sample prepared without NaF showed a pure anatase phase, while the NaF sample successfully showed a pure brookite phase. It is worth noting that the concentration of NaF was an important factor, and the pure brookite phase only appeared with a Na: Ti molar ratio of 1:1 or more.

For the fourth sample as shown in Figure 9, the pure nanoparticles (TiO_2_-NPs) form in the absence of a morphology controlling agent (alcoholysis of only TNBT in ethanol), as is the case with the results from TEM.

The calculated crystallite sizes according to the Scherrer formula for TiO_2_-NSs, TiO_2_-NRsand TiO_2_-NPs are shown in Table 2. It can be seen that the sample of TiO_2_-NSs has larger average crystal size and better crystallinity. This is consistent with the results of TEM presented above.

## 7. Optical Properties

Figure 10 shows the optical transmittance spectra (T) of TiO_2_-NCs films deposited on FTO glass substrates. They show the optical transmittance (T) in the visible range for films with a thickness of 10 m (measured with a surface profilometer). High absorption is visible when the wavelength is less than 400 nm. The transmittance of the films is less than 30% and increases almost linearly with increasing wavelength. As the wavelength decreases, the absorbance and absorption coefficient (α) values begin to decrease the transmittance until the region of strong absorption is reached; here, the transmittance decreases dramatically, almost entirely due to the influence of α. Interference-free transmittance and reflectance spectra characterize the region of strong absorption. Figure 10 depicts the optical transmission findings. The data are normalized to 100% transmission at 900 nm, where brookite is transparent.

UV/Vis spectroscopy of TiO_2_-NCs was performed to evaluate the effects of quantum size. Figure 11a shows the strong band edge absorption of TiO_2_-NCs at 390 nm, which has a larger blue shift than TiO_2_. Based on the absorption spectrum of TiO_2_-NCs, the excitonic or interband (valence conduction band) transition can be resolved, allowing us to calculate the band gap energy.

We measured the optical absorbance edge of anatase and brookite TiO_2_ at room temperature. The brookite observed edge is wide and stretches throughout the visible, in contrast to anatase’s steep edges. There is no proof of a direct gap up to about 4.5 eV. As a result, we classify anatase and brookite TiO_2_ as an indirect-gap semiconductor with a room-temperature operating range.

By analyzing the optical absorption spectra and applying the following formula (3) [48,49,50], the direct and indirect transitions in the band gap of the material can be determined.
αhν = B(hνـE_g_)^n^; α = 2.303 (A/d) (3)
where A and d are the absorptivities and the film thickness, respectively, h is Planck’s constant, the frequency of the incident radiation, B is a constant that depends on the transition probability, E_g_ is the energy band gap, and n = 1/2 and 2 for direct and indirect band gaps, respectively [51]. The existence of both direct and indirect band gaps is confirmed by the linearity in the plots of (*α*h*ν*)^2^ and (*α*h*ν*)^1/2^ as a function of photon energy (h*ν*), for direct and indirect band gaps, respectively (see Figure 7). By extrapolating the linear component of (αhν)^1/2^ versus hν, where an is the absorption coefficient and h is the photon energy [52,53,54], the apparent optical bandgap energy of TiO_2_ NCs was estimated. As a result, the band gap energies (Figure 11) for anatase-TiO_2_-NSs (Sample 1) and brookite-TiO_2_-NRs (Sample 3) were calculated to be 2.85 and 3.08 eV, respectively.

In general, compared to rutile and brookite, the anatase crystalline phase exhibits much higher photocatalytic activities [20]. Because photogenerated electrons cannot directly transition from the conduction band (CB) to the valence band (VB) of anatase TiO_2_, indirect band gap anatase exhibits a longer lifetime of photoexcited electrons and holes than direct band gap brookite. The spectral dependence of absorption (Figure 10 and Figure 11) strongly indicates that brookite TiO_2_ is an direct-gap semiconductor with a larger indirect-gap than anatase TiO_2_ [20,55]. However, both types have a large band gap, it can absorb several percent of sunlight in the near-UV region [1,2,3].

## 8. Conclusions

In summary, TiO_2_-NCs with exposed high-energy {001} facets were prepared by a facile HF -mediated and NaF-mediated hydrothermal method. The growth of nanocrystals with exposed high-energy facets is of particular importance due to the high reactivity of these facets. Anatase TiO_2_ nanosheets can be prepared by the hydrothermal method in the presence of HF with a width of 20–30 nm and a thickness of 5–7 nm could be grown, corresponding to about 80% of the {001} outer surface in the nanocrystal. Ti(OBu)_4_ alone was subjected to alcoholysis in ethanol, yielding only pure anatase TiO_2_ nanoparticles in absence of HF. In addition, this work succeeded in generating a high-purity brookite TiO_2_ NRs structure, which is the most difficult TiO_2_ polymorph to synthesize. The brookite TiO_2_ nanorods were hydrothermally synthesized using NaF as the fluorine source required for {001} facet formation. By using NaF, other methods were developed in this work to replace HF as a hazardous chemical. Additionally, transmission results confirmed both types of anatase and brookite have a large band gap, and they can absorb several percentages of sunlight in the near-UV region. Finally, these nanostructures will be investigated for applications in dye-sensitized solar cells (DSSCs) in our future work. In addition to its possible use in DSSCs, TiO_2_ is also incorporated into the structure of perovskite solar cells, which hold great promise. The fabricated TiO_2_ nanosheet electrode is also of great interest for photocatalysis, catalysis, electrochemistry, separation, purification, etc.

## Data Availability

The data presented in this study are available on request from the corresponding author.

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
