# Peer review of "Successful Growth of TiO2 Nanocrystals with {001} Facets for Solar Cells"

_nanomaterials, 2023, doi:10.3390/nano13050928_

Round 1
Reviewer 1 Report
This maniscript discusses the use of non-aqueous solvothermal methods to synthesize TiO2 nanostructures, specifically anatase nanosheets and brookite nanorods, using tetrabutyl titanate Ti(OBu)4 as a precursor and hydrofluoric acid (HF) and sodium fluoride (NaF) as morphology control agents. The study investigates the impact of using fluorine as a morphology control agent on the structure and morphology of TiO2 nanostructures, as well as the resulting impact on the photoelectric conversion efficiency of dye-sensitized solar cells (DSSCs). The authors used various analytical techniques, including TEM, HRTEM, and XRD, to characterize the structure and morphology of the synthesized TiO2 nanostructures. The results show that the use of fluorine can enhance the photocatalytic activity and control the particle morphology of TiO2, leading to improved photoelectric conversion efficiency in DSSCs. The study provides insights into the synthesis of TiO2 nanostructures using non-aqueous solvothermal methods and their potential application in DSSCs.
I think this manuscript has a potential impact on the field of materials science and could be a valuable contribution to the design of new metal oxides for solar cells. Overall, this manuscript represents a contribution to the field and is recommended for publication in the Nanomaterials with the following minor revisions. Prior to publication, authors should address the following minor issues listed below:
1) The linear regression equation for the data in Figure 6 should be provided.
2) Related references should be added in the introduction.
a. Lu and coworkers reported the synthesis of anatase TiO2 microcrystals with high-energy {001} facets by using hydrofluoric acid as a shape-controlling agent.
b. Han and coworkers used a similar strategy to synthesize anatase TiO2 nanosheets with 89% exposed {001} facets using a hydrofluoric acid solution as a solvent.
3) In Figure 5, Figure 7, Figure 8 and Figure 9, the abscissa needs to maintain a consistent format and units.
4) In Figure 1, Figure 2 and Figure 4, some letters used for sorting are missing.
5) The scale bar of Figure 4b is too low resolution to be seen clearly.
6) The authors should add some literature descriptions to make the manuscript more convincing. I would like to suggest the authors cite the following relevant articles:
a. Nanomaterials 2022, 12(9), 1589 High-Efficiency Crystalline Silicon-Based Solar Cells Using Textured TiO2 Layer and Plasmonic Nanoparticles
b. Nanomaterials 2023, 13(1), 186 Enhancement of Perovskite Solar Cells by TiO2-Carbon Dot Electron Transport Film Layers
c. Nano Research Energy 2022, 1: e9120020 Recent research progress on operational stability of metal oxide/sulfide photoanodes in photoelectrochemical cells
d. Nanomaterials 2022, 12(7), 1165 Manufacturing a TiO2-Based Semiconductor Film with Nanofluid Pool Boiling and Sintering Processes toward Solar-Cell Applications
Author Response
Manuscript Number: nanomaterials-2257492
" Successful Growth of TiO2 nanocrystals with {001} facets for Solar Cells "
by Saif Qaid and et al.
  **********************************************************************
Responses to Reviewer #1:
Reviewer #1: Comments and Suggestions for Authors
This maniscript discusses the use of non-aqueous solvothermal methods to synthesize TiO2 nanostructures, specifically anatase nanosheets and brookite nanorods, using tetrabutyl titanate Ti(OBu)4 as a precursor and hydrofluoric acid (HF) and sodium fluoride (NaF) as morphology control agents. The study investigates the impact of using fluorine as a morphology control agent on the structure and morphology of TiO2 nanostructures, as well as the resulting impact on the photoelectric conversion efficiency of dye-sensitized solar cells (DSSCs). The authors used various analytical techniques, including TEM, HRTEM, and XRD, to characterize the structure and morphology of the synthesized TiO2 nanostructures. The results show that the use of fluorine can enhance the photocatalytic activity and control the particle morphology of TiO2, leading to improved photoelectric conversion efficiency in DSSCs. The study provides insights into the synthesis of TiO2 nanostructures using non-aqueous solvothermal methods and their potential application in DSSCs.
I think this manuscript has a potential impact on the field of materials science and could be a valuable contribution to the design of new metal oxides for solar cells. Overall, this manuscript represents a contribution to the field and is recommended for publication in the Nanomaterials with the following minor revisions. Prior to publication, authors should address the following minor issues listed below:
We thank the reviewer for his positive attitude towards our results, and we thank him also for the valuable comments and suggestions, the responses to which are as follows:
In the revised version, all additions and corrections were highlighted in green color.
******************************************************************************
******************************************************************************
- The linear regression equation for the data in Figure 6 should be provided.
We thank the reviewer for his suggestion.
The linear regression equation for the data in Figure 6 was provided and the text and Figure 6 were updated.
The correction is done and this is now explicitly mentioned in the revised manuscript.
******************************************************************************
- Related references should be added in the introduction.
- Lu and coworkers reported the synthesis of anatase TiO2 microcrystals with high-energy {001} facets by using hydrofluoric acid as a shape-controlling agent.
- Han and coworkers used a similar strategy to synthesize anatase TiO2 nanosheets with 89% exposed {001} facets using a hydrofluoric acid solution as a solvent.
We thank the reviewer for his comment.
Both references were added in the revised manuscript.
Also, the references section was revised.
******************************************************************************
- In Figure 5, Figure 7, Figure 8 and Figure 9, the abscissa needs to maintain a consistent format and units.
We thank the reviewer for his comment.
The abscissa in the above figures maintained a consistent format and units.
******************************************************************************
- In Figure 1, Figure 2 and Figure 4, some letters used for sorting are missing.
We thank the reviewer for his comment.
The figures were updated.
******************************************************************************
- The scale bar of Figure 4b is too low resolution to be seen clearly.
We thank the reviewer for his comment.
The scale bars were rewritten in the caption for more clarity.
******************************************************************************
- The authors should add some literature descriptions to make the manuscript more convincing. I would like to suggest the authors cite the following relevant articles:
- Nanomaterials 2022, 12(9), 1589 High-Efficiency Crystalline Silicon-Based Solar Cells Using Textured TiO2 Layer and Plasmonic Nanoparticles
- Nanomaterials 2023, 13(1), 186 Enhancement of Perovskite Solar Cells by TiO2-Carbon Dot Electron Transport Film Layers
- Nano Research Energy 2022, 1: e9120020 Recent research progress on operational stability of metal oxide/sulfide photoanodes in photoelectrochemical cells
- Nanomaterials 2022, 12(7), 1165 Manufacturing a TiO2-Based Semiconductor Film with Nanofluid Pool Boiling and Sintering Processes toward Solar-Cell Applications
Thank you for the comment.
Based on your request the above papers were added as a reference in the revised manuscript in their appropriate locations add some literature descriptions.
Also, the references section was revised.
******************************************************************************
**********************************************************************

Reviewer 2 Report
Manuscript ID: nanomaterials-2257492
Title: Successful Growth of TiO2 nanocrystals with {001} facets for Solar Cells
This manuscript reports the synthesis of titanium dioxide (TiO2) nanostructures and their characterization by TEM, HRTEM, SAED, and TEM. The authors investigated the synthesis and characterization of single-crystal anatase TiO2 nanosheets and brookite TiO2 nanorods. This manuscript has scientific soundness as these materials are of great interest to diverse areas of research and technologies. Nevertheless, I think that the scientific content of the paper could be improved by the authors and some issues should be clarified before considering this paper for publication:
11) As a general comment I do think that although the results seem to be quite good and well described, some parts of the manuscript seem like a report instead of a scientific article. I suggest improving the discussion in order to have more strong conclusions and whenever possible support the ideas with literature data.
22) Some parts of the manuscript are too repetitive. For instance: lines 45 and 45 and lines 68 and 69 of the introduction section. Please check all the parts of the manuscript.
3) Abstract: “…TiO2-NSS with an average area of about 20-30 nm…”. Please correct the sentence, the values do not correspond to an area.
44) The authors started the introduction with a description of the potential application of TiO2 for dye-sensitized solar cells. I would recommend also mentioning the perovskite solar cells, very promising and also incorporate TiO2 in their architecture. The authors can use, for instance, the following references as support: “ACS Nano 2015, 9, 1, 564–572; Journal of Power Sources 286 (2015) 118–123; Solar Energy Materials & Solar Cells 149 (2016) 1–82; Thin Solid Films 664 (2018) 12–18; Energies 2021, 14(23), 7870”.
55) What are the main reasons for rutile being the most stable in the bulk phase whereas anatase is more stable at the nanoscale? Some comments would be helpful.
66) Despite the difficulty of creating brookite nanostructures, the authors have described in their work the creation of brookite TiO2 nanorods. Please highlight the main difficulties regarding the synthesis of brookite.
77) If available, please add literature data to support the experimental method used for the preparation of brookite nanorods.
88) Please update Figure 1. The caption (a) is overlapped with the Figure.
99) I recommend the authors present additional details relating to the method for deriving the planes of anatase 001 from the lattice spacing (~0,35).
110) What is the main role of NaF concentration on the morphology and size of TiO2-NRs?
111) The authors derive the band gap by using the Tauc plot. Please include the following references to validate this method: Optical Materials 58 (2016), 51-60; Applied Physics B 119 (2015), 273-279; Journal of solid-state chemistry 240 (2016), 43-48.
112) Is there any specific reason for the obtained differentiation between the band gaps of brookite and anatase?
113) In the conclusion section, I would advise the authors to make much clearer the findings of this work and their potential impact on the development of solar cells.
Best regards.
Author Response
Manuscript Number: nanomaterials-2257492
" Successful Growth of TiO2 nanocrystals with {001} facets for Solar Cells "
by Saif Qaid and et al.
  **********************************************************************
Responses to Reviewer #2:
Reviewer #2: Comments and Suggestions for Authors
This manuscript reports the synthesis of titanium dioxide (TiO2) nanostructures and their characterization by TEM, HRTEM, SAED, and TEM. The authors investigated the synthesis and characterization of single-crystal anatase TiO2 nanosheets and brookite TiO2 nanorods. This manuscript has scientific soundness as these materials are of great interest to diverse areas of research and technologies. Nevertheless, I think that the scientific content of the paper could be improved by the authors and some issues should be clarified before considering this paper for publication:
We thank the reviewer for his positive attitude towards our results, and we thank him also for the valuable comments and suggestions, the responses to which are as follows:
In the revised version, all additions and corrections were highlighted in green color.
******************************************************************************
******************************************************************************
- As a general comment I do think that although the results seem to be quite good and well described, some parts of the manuscript seem like a report instead of a scientific article. I suggest improving the discussion in order to have more strong conclusions and whenever possible support the ideas with literature data.
Thank you for the suggestion.
We try to improve the discussion based on your suggestion.
******************************************************************************
- Some parts of the manuscript are too repetitive. For instance: lines 45 and 45 and lines 68 and 69 of the introduction section. Please check all the parts of the manuscript.
We are sorry for this mistake.
All typos were corrected throughout the whole manuscript.
******************************************************************************
- Abstract: “…TiO2-NSS with an average area of about 20-30 nm…”. Please correct the sentence, the values do not correspond to an area.
The correction is done in the revised manuscript.
******************************************************************************
- The authors started the introduction with a description of the potential application of TiO2 for dye-sensitized solar cells. I would recommend also mentioning the perovskite solar cells, very promising and also incorporate TiO2 in their architecture. The authors can use, for instance, the following references as support: “ACS Nano 2015, 9, 1, 564–572; Journal of Power Sources 286 (2015) 118–123; Solar Energy Materials & Solar Cells 149 (2016) 1–82; Thin Solid Films 664 (2018) 12–18; Energies 2021, 14(23), 7870”.
The required has been done in the revised manuscript. So, some sentences were added to start the introduction section.
Based on your request the above papers were added as a reference in the revised manuscript in their appropriate locations add some literature descriptions.
Also, the references section was revised.
******************************************************************************
- What are the main reasons for rutile being the most stable in the bulk phase whereas anatase is more stable at the nanoscale? Some comments would be helpful.
Of course, the rutile is the most stable in the bulk phase. While the anatase and brookite are metastable in the bulk phase.
The key to boosting future advances in all promising technological areas that benefit from nanosized TiO2 resides in the ability to develop nanostructures fulfilling several requirements, such as defined crystal phases and degree of crystallinity, engineered dimensions and shapes, and suitable chemical functionalities at the surface.
So, the anatase becomes more stable at the nanoscale but not the most.
******************************************************************************
- Despite the difficulty of creating brookite nanostructures, the authors have described in their work the creation of brookite TiO2 Please highlight the main difficulties regarding the synthesis of brookite.
The required has been done. The main difficulties regarding the synthesis of brookite were added to the revised manuscript.
******************************************************************************
- If available, please add literature data to support the experimental method used for the preparation of brookite nanorods.
To the best of my knowledge, we have not been able to obtain previous reports that prepared brookites as is the case in this work. But some experimental method used for the preparation of brookite nanosized was added.
******************************************************************************
- Please update Figure 1. The caption (a) is overlapped with the Figure.
The required has been done.
******************************************************************************
- I recommend the authors present additional details relating to the method for deriving the planes of anatase 001 from the lattice spacing (~0,35).
Thank you for the comment.
This corresponded to the distance between adjacent {001} lattice planes, which was confirmed by performing XRD analysis from D values at separated peaks.
******************************************************************************
- What is the main role of NaF concentration on the morphology and size of TiO2-NRs?
Thank you for the comment.
The ratio of brookite and anatase TiO2 can be tailored by the NaF concentration. In a concentrated NaF solution, high-quality brookite TiO2 was acquired. the nominal atomic ratio of fluorine to titanium was 1 because fluorine can enhance the photocatalytic activity and control the particle morphology of TiO2. So, some sentences were added to start the introduction section.
******************************************************************************
- he authors derive the band gap by using the Tauc plot. Please include the following references to validate this method: Optical Materials 58 (2016), 51-60; Applied Physics B 119 (2015), 273-279; Journal of solid-state chemistry 240 (2016), 43-48.
Thank you for the comment.
Based on your request the above papers were added as a reference in the revised manuscript in their appropriate locations add some literature descriptions.
Also, the references section was revised.
******************************************************************************
- Is there any specific reason for the obtained differentiation between the band gaps of brookite and anatase?
Thank you for the comment.
Indirect band gap anatase exhibits a longer lifetime of photoexcited electrons and holes than direct band gap brookite because the direct transitions of photogenerated electrons from the conduction band (CB) to the valence band (VB) of anatase TiO2 are impossible.
******************************************************************************
- In the conclusion section, I would advise the authors to make much clearer the findings of this work and their potential impact on the development of solar cells.
Thank you for the comment.
The correction is done.
Now the paragraph of the " concluding " section was revised and rewritten in some sentences for more clarity as shown in the revised manuscript. The conclusions were improved to meet your points put forward. Also, the manuscript text is restructured again at that point in the revised manuscript.
******************************************************************************
******************************************************************************

Reviewer 3 Report
see attachment

Author Response
Manuscript Number: nanomaterials-2257492
" Successful Growth of TiO2 nanocrystals with {001} facets for Solar Cells "
by Saif Qaid and et al.
  **********************************************************************
Responses to Reviewer #3:
Reviewer #3: Comments and Suggestions for Authors
Dear Editor,
The authors present a study on obtaining some nanostructures based on TiO2 with exposed {001} facets for applications in solar cells and catalysts. The work is of scientific interest, but before it can be published, a series of improvements are necessary:
We thank the reviewer for his positive attitude towards our results, and we thank him also for the valuable comments and suggestions, the responses to which are as follows:
In the revised version, all additions and corrections were highlighted in green color.
******************************************************************************
******************************************************************************
- Thorough checking of the text to correct drafting errors;
Thank you for the comment.
The corrections are done.
The language was polished and revised. It was also proofread again in the revised manuscript.
******************************************************************************
- For the uniformity of the text, I recommend that some improvements be made in the abstract, This project; in this work; etc.
Thank you for the comment. The uniformity of the text was checked in the abstract.
******************************************************************************
- It is recommended to give the unit of measure for, Molar mass=340.32;
The Molar mass unit was given. The correction is done and this is now explicitly mentioned in the revised manuscript.
******************************************************************************
- For the Scherrer equation, it is recommended to give references;
The required has been done.
******************************************************************************
- In Figure 5, Miller indices should be in round brackets;
The required has been done.
******************************************************************************
- Also for uniformity, the names of the axes in Figure 9 should keep the same format as in the previous ones.
The required has been done.
******************************************************************************
- The numbering of equations;
The required has been done.
******************************************************************************
- The conclusions are recommended to be more concise, highlighting the properties that would lead to the successful application of these materials.
Thank you for the comment.
Now the paragraph of the " concluding " section was revised and rewritten in some sentences for more clarity as shown in the revised manuscript. The conclusions were improved to meet your points put forward. Also, the manuscript text is restructured again at that point in the revised manuscript.
******************************************************************************
- References are recommended to be written according to the established format,
[3] Hoang and et al S. Chemical bath
[4] k. Kalyanasundaram and et al. Dye
[30], et al. Co2+ Doping and Molecular Adsorption Behavior of Anatase TiO2 (001) Crystal Plane. Catal Res 2022;2:1–1 etc.
Thank you for the comment.
The correction is done.
Also, the references section was revised, and make the missing data.
******************************************************************************
******************************************************************************

Round 2
Reviewer 2 Report
The authors tried to answer all the questions/comments comprehensively. In my opinion, I believe the manuscript might be suitable for publication.
Regards.